# Discrimination/Classification of Edible Vegetable Oils from Raman Spatially Solved Fingerprints Obtained on Portable Instrumentation

**DOI:** 10.3390/foods13020183

**Published:** 2024-01-05

**Authors:** Guillermo Jiménez-Hernández, Fidel Ortega-Gavilán, M. Gracia Bagur-González, Antonio González-Casado

**Affiliations:** 1Department of Analytical Chemistry, Faculty of Science, University of Granada, C/Fuentenueva w/n, 18071 Granada, Spain; gjh188@correo.ugr.es (G.J.-H.); agcasado@ugr.es (A.G.-C.); 2Animal Health Central Laboratory (LCSA), Department of Chemical Analysis of Residues, Ministry of Agriculture, Fisheries and Food, Camino del Jau w/n, 18320 Santa Fe, Spain

**Keywords:** Raman fingerprints, edible vegetable oils, sunflower oil, olive oil, pattern recognition techniques, portable analyser, SORS

## Abstract

Currently, the combination of fingerprinting methodology and environmentally friendly and economical analytical instrumentation is becoming increasingly relevant in the food sector. In this study, a highly versatile portable analyser based on Spatially Offset Raman Spectroscopy (SORS) obtained fingerprints of edible vegetable oils (sunflower and olive oils), and the capability of such fingerprints (obtained quickly, reliably and without any sample treatment) to discriminate/classify the analysed samples was evaluated. After data treatment, not only unsupervised pattern recognition techniques (as HCA and PCA), but also supervised pattern recognition techniques (such as SVM, kNN and SIMCA), showed that the main effect on discrimination/classification was associated with those regions of the Raman fingerprint related to free fatty acid content, especially oleic and linoleic acid. These facts allowed the discernment of the original raw material used in the oil’s production. In all the models established, reliable qualimetric parameters were obtained.

## 1. Introduction

Sunflower oil from the seeds of *Helianthus annuus* L. [1] is produced in several countries, including Russia, Ukraine and China. In the European Union, it represents the second most important source of seed oils after rapeseed oil; the area under sunflower crops will increase up to 1.0 million ha by the end of 2031 [2]. The plant grows best in dry, temperate climates (with an average temperature of 20–25 °C) with high solar radiation, low humidity, and deep soils to spread its roots in search of nutrients and water. Its deshelled seeds are responsible for 80% of the total fruit weight and have a high oil content up to 55% *w*/*w* [3,4]. These oils have a high lipid content, most of which are triglycerides composed mainly of long-chain unsaturated fatty acids of different unsaturation, with linoleic acid (59%, polyunsaturated omega 6) being the most important [5,6]. In addition, the presence of other fatty acids such as oleic acid (30%, monounsaturated omega 9), stearic acid (6%, saturated), and palmitic acid (5%, saturated) should be noted.

Like olive oil (OO), there are several commercial categories of sunflower oil, the main ones being as follows: (i) refined sunflower oil (SFO), characterised by a high linoleic acid (LA) content; (ii) high oleic sunflower oil (HOSFO), with an oleic acid (OA) content of at least 75% measured as a percentage of the total fatty acid content; and (iii) medium oleic sunflower oil (MOSFO), with a seed OA content ranged from 50% to 75%. The last two commercial categories come from seeds genetically modified to naturally increase not only the ratio of oleic to linoleic acid but also other monounsaturated fatty acids or vitamins such as vitamin E. These facts confer to these oils, especially to HOSFO, an overall composition with remarkable similarity to OO, and as an extra factor, greater resistance to oxidation and increased possibilities for use [7,8].

Currently, the main analytical methods used to check authenticity and evaluate possible adulteration of edible vegetable oils, especially OO, are chromatography-based analyses of the presence of triglycerides and fatty acids [9,10]. The most used are Thin-Layer Chromatography (TLC), High-Pressure Liquid Chromatography (HPLC), and Gas Chromatography (GC). TLC is used both qualitatively and quantitatively in official methods such as the analysis of sterol and stanol fractionation in oils [11,12]. HPLC is, among other uses, an International Olive Council reference method commonly used in routine laboratories [13,14]. GC is used not only as the GC-FID (Flame Ionization Detector) official method specified in IOCs guides, but also for characterising the triglyceride profile in vegetable oils, or for the separation and quantification of fatty acids according to their esterified fraction (FAME) by means of a prior hydrolysis and methylation of fatty acids step [15,16,17].

As an alternative to chromatographic techniques, the spectroscopic ones (Raman, MIR, FIR, NMR, etc.) provide simple, fast and reliable results, with the advantage of being used directly on the sample without the need for any sample pre-treatment stage. Moreover, they appear to be not only economical but also environmentally sustainable analytical methods [17,18,19]. NMR provides a quick way of measuring the oil content of oil seeds such as sunflower seeds, as well as assessing their resistance to high temperatures by evaluating the degradation of their constituents in their main domestic use, i.e., food frying [20]. Some of these techniques, such as Raman or NIR spectroscopy, can be easily adapted to portable devices, allowing the acquisition of analytical data in situ in real time, although in many cases these techniques have proven to be incapable of measuring through packaging. To overcome this inconvenience, some instrumental improvements have been made to Raman spectroscopy, as in the case of the Spatially Offset Raman spectroscopy (SORS) technique [21]. Despite these advantages, they have a lower resolution capability than chromatographic techniques, as they do not provide information on each individual compound present, but rather on the bonds that occur in the components of a sample as a whole.

Considering that the final output of an analytical device (bench top, handheld or portable) is an analytical signal, the use of this to obtain information about the properties of a material related to its chemical composition, e.g., fingerprinting methodology, has shown to be a powerful tool to identify, discriminate and authenticate edible oils among other commodities. In this sense, these signals may relate to the identity of a food, to its physico-chemical or other natural properties, as well as to the presence or quantity of compounds in its chemical composition [22,23]. Despite this, instrumental fingerprints contain hidden information, which normally requires the use of chemometric tools in particular, or machine learning methods in general. Multivariate data analysis is essential for handling statistically complex data such as Raman fingerprints. The extraction of valuable information from large and complex chemical measurement data has become a significant research area. Chemometrics and machine learning are interdisciplinary methods that integrate chemistry, mathematics, computer science, and other fields to solve data-processing problems in modern analytical instruments [24,25,26].

In the food sector, fingerprinting methodology has been widely used to solve different problems, such as the authentication and discrimination of olive oils, margarines, and fat spread, or the evaluation of the quality and production process of spirits, among other foodstuffs [27,28,29,30,31]. Regarding the use of handheld or portable devices based on SORS applications, few studies have been carried out despite the great potential of the technique [21]. Recent examples include the detection of possible adulteration in alcoholic beverages [32]; the study of the evolution of the alcoholic fermentation process in white wine [33]; the authenticity assessment of the origin of animal milk used in the production of cheese; the characterisation of several commercial categories of cheese (Cheddar, Manchego and Pecorino Romano); the analysis of packaged margarines and fat spreads [34,35,36]; and the study of the adulteration of extra virgin olive with other edible vegetable oils, as in the case of Varnasseri et al. [37].

Thus, the aim of this work is to propose a more environmentally friendly and sustainable analytical method that combines the Raman fingerprints obtained from a portable instrument with chemometrics/machine learning, to build classification models to reliably differentiate sunflower oils from olive oils of different commercial categories.

## 2. Materials and Methods

### 2.1. Oil Samples

A sample bank consisting of 145 samples from different types of edible vegetable oils, purchased at local supermarkets, was used for this study; these were separated into two main groups: sunflower and olive oil. The first group was constituted by 48 samples where 7 of them were labelled as high oleic acid content. The second group was composed of a total of 97 samples from which 65 were extra virgin olive oil (EVOO), 22 were virgin olive oil (VOO), and 10 were pomace olive oil (POO).

### 2.2. Spectroscopic Analysis

A Vaya Raman portable spectrometer (Agilent Technologies, Santa Clara, CA, USA) was used in this study. It was equipped with a 3B class diode laser operating at 830 nm and a maximum power of 450 mW. The exposure time of the sample to the laser varied between 0.5 and 2 s. The spectral resolution of the device was 12–20 cm^−1^, the wavenumber range between 735–1540 cm^−1^, and a cooled CCD (charge-coupled device) was used as a detection system. The specific offset length from the incidence point was fixed by equipment at 0.6 cm. For measurement (with a total measurement time between 30 s and 2 min), edible oil samples were introduced into 4 mL borosilicate glass vial of 1mm thickness.

After carrying out the measurement, the equipment software performed the following: (i) a correction from the information was obtained to eliminate any possible influence of the container, (ii) a baseline adjustment, and finally (iii) a normalisation of the intensity values. Therefore, the result of each analysed sample was a Raman spectrum from 350 to 2000 cm^−1^ in which the intensity values were between 0 and 1. Thus, the final spectrum was a normalised spatially resolved Raman spectrum (NSR Raman Spectrum).

### 2.3. Data Treatment

Each NSR RAMAN spectrum was collected in .CSV format (comma separated value) and exported to .mat format for the subsequent elaboration of the fingerprints matrix within the MATLAB environment (version 9.3, Mathworks Inc., Natick, MA, USA). As a result, a comprehensive 145 × 1651 data matrix consisting of 145 NSR RAMAN fingerprints (rows), each of which was composed of 1651 normalised intensity values (columns), was obtained as a raw data matrix and once pre-processed, used for the different chemometric studies.

The preprocessing stage included the following: a selection of interval of interest for each fingerprint; a filtering smoothing of the signals using a Savitzky-Golay filter (1st derivate, 2nd order polynomial and a filter width of 21 points); and a mean centre, obtaining a reduced-preprocessed NSR RAMAN fingerprints matrix of 145 samples × 902 variables. Figure 1 shows the NSR RAMAN fingerprint of edible vegetable oils analysed for this study before and after the pre-processing step. Unsupervised and supervised pattern recognition techniques were explored using PLS_Toolbox, version 8.6.1 (Eigenvector Research Inc., Manson, WA, USA) [38].

## 3. Results and Discussion

### 3.1. Unsupervised Pattern Recognition Methods

The analysis of natural grouping trends could allow for establishing a correlation between the data in the NSR RAMAN fingerprints and their impact on the vegetable origin of the edible oils analysed. This would lead to evaluating the use of green analytical technology such as a ‘Vaya Raman’ portable spectrometer, to distinguish between vegetable oils obtained from different types of raw materials such as olive and sunflower.

#### 3.1.1. Hierarchical Cluster Analysis of Raman Edible Vegetable Oil Fingerprints

An HCA was performed using the previously defined data matrix. Ward’s method and Manhattan distance were used as the linkage criterion and measure of distance between pairs of observations, respectively. The number of clusters was selected using Dlinkage = 2/3 of Dmax as the internal criterion. The edible oils clustered naturally, based on the raw material used and the oleic acid content. Thus, two main clusters could be observed. Group I was constituted by oils obtained from sunflower seeds, while group II included all the olive oils (extra virgin, virgin, and pomace) and high oleic sunflower oils (Figure 2).

Regarding group I, it also could be observed that a previous partial clustering took place according to the percentage of oleic acid shown on the label of the commercial container, where natural nesting distinguished those sunflower oils with oleic acid, locating them in the upper part of the group. Although a similar behaviour could be observed when group II was analysed, in this case the nesting order was the opposite to the previous one, due to the upper partial cluster groups of olive oil samples with low acidity, which implied a lower free oleic acid content. Meanwhile, those samples with a higher content of this acid appear grouped in the second partial cluster (some EVOO, refined and pomace samples). Thus, the appearance of those sunflower oil samples (commercially labelled as high oleic acid content sunflower oils) in this last sub-group could be justified.

#### 3.1.2. Principal Component Analysis

When PCA was applied to the data matrix, five principal components (PCs) were obtained which explained 92.42% of the variance of the model for the edible vegetable oils analysed. PC1 explained 88.99% of the total variance of the system, while the other four principal components explained the remaining 3.43% of the variance (PC2: 1.06%; PC3 0.94%; PC4: 0.73% and PC5: 0.70%).

In Figure 3a, the scores received by the analysed oil samples in the first two components (PC2 vs. PC1) are illustrated. The edible vegetable oils are grouped according to the raw material used in the elaboration process, similar to what occurs when HCA is applied. Most of the sunflower oils received negative scores for PC1 (Group I), while the olive oils scored positively for this component (Group II). Additionally, the high oleic sunflower oils included in this group received negative scores for PC2.

Figure 3b presents a graphical representation of the scores received by the analysed oil samples in the PC5 vs. PC1 space. The distribution of the edible oils’ scores along the PC1 space reveals the same main grouping as the raw material from which the oils are obtained (Group I and II). Additionally, a new tendency is observed in Group II, which can be explained by the oleic acid content. The oils with a high oleic acid content received positive scores on PC5, while the others were in the negative area of this component. In addition, it was observed that higher oleic acid content in the oil samples resulted in higher positive scores on PC1. These trends can be explained by examining the fingerprint regions with minimal variations in total variance, indicating minimal differences between fingerprints.

When the 3D space is defined as PC5 vs. PC2 vs. PC1 (Figure 3c), a clearer grouping can be observed. Thus, sunflower oils received increasingly positive PC1 scores, as the oleic acid content was higher. In addition, the olive pomace oils received positive values from PC1, PC2, and PC5, which could allow it to distinguish between sunflower, virgin/extra virgin olive oils, and olive pomace oils. These results are consistent with those found by several authors, indicating that olive oil and sunflower oil can be distinguished regardless of whether the study is to identify adulterations or to authenticate samples [39,40]. However, it is worth noting that when the oleic acid content is taken into account, sunflower oil can resemble olive oil in cases where the seed has been genetically modified to have this characteristic.

If the loading plots of each component (Figure 4a–c) are analysed, based on Figure 4a, it can be observed that the PC1 loadings would allow the identification of those variables that explain the natural grouping according to the raw material. The variables (wave numbers) associated with the bands in “conventional” Raman spectra associated with free fatty acid (mainly oleic and linoleic acids) [41] are as follows: (i) v(C=C) vibration of *cis*-alkene, stretching of double bond (≈1650 cm^−1^); (ii) δ(C–H) twisting of CH_2_ (≈1300 cm^−1^); (iii) δ(=C–H) scissoring (≈1250 cm^−1^); (iv) v(C–C) and δ(C=C) twisting of *trans*-alkene (≈1000 cm^−1^); and (v) v(C–H) symmetric and/or asymmetric stretching (≈800 cm^−1^). Regarding Figure 4b, it can be observed that for the PC2 loadings the wavenumbers in which the free fatty acid information appears are as follows: (i) v(C=C) vibration of *cis*-alkene (≈1650 cm^−1^); and (ii) v(C–H) symmetric and/or asymmetric stretching (≈800 cm^−1^), the first one being the most influential in the natural grouping observed. When the PC5 loadings are considered (Figure 4c), the natural grouping could once again be associated with the same wavenumbers in PC2. The main difference between PC2 and PC5 loadings arose from the v(C–H) symmetric and/or asymmetric stretching (≈800 cm^−1^). Thus, slight variations on these Raman modes provoke the separation between the edible olive oils analysed. This is consistent with previous publications that have characterised the main Raman bands of EVOO and SFO, where the different vibrational modes of C=C and C–C can distinguish between the spectra of EVOO and SFO. This fact confirms that Raman spectroscopy is a reliable analytical technique not only for adulteration studies, as Philippidis et.al. quoted in 2017, but also for authentication purposes [42,43,44,45].

### 3.2. Supervised Pattern Recognition Methods

In order to study the capability of the NSR RAMAN fingerprint to discriminate/classify between the edible vegetable oils analysed, several one-input class classification/discrimination models, as SVM, kNN and SIMCA, were developed. For these purposes, the target class was assigned to sunflower oils, with the not-SF (olive oils) considered the non-target class. The two necessary sets (calibration and prediction) were selected using the Kennard–Stone algorithm [38,39,46], keeping 66.6% of the original samples for calibration/cross-validation purposes and the remaining 33.4% for external prediction purposes.

The SVM model was developed without reducing the data dimensionality (uncompressed). The Kernel algorithm used a radial basis function (RBF) with the default values for gamma and cost parameters in the PLS_Toolbox™.

It can be seen in Figure 5 how only one sample labelled as sunflower oil was classified as not-SF. In addition, other samples labelled as Sunflower appeared closer to the threshold, which should be considered as badly classified despite appearing in the sunflower prediction probability area. Regarding the information included in the commercial label of the analysed samples, all of them appear described as high oleic sunflower oils with different content percentages of this acid, which explains the behaviour illustrated in this Figure.

The discrimination/classification power of the NSR RAMAN fingerprint was evaluated using neighbour distances. A kNN with k = 7 was found to be the best value for deciding the neighbour distance in the model. The sunflower class was defined by a predicted probability value of 1, while the non-target class (olive oil) was defined by a probability of 0. Considering the two hard modelling techniques used, a similar behaviour can be observed, the main difference being the probability prediction value belonging to each class. All the sunflower samples labelled without high oleic acid content were well classified (probability = 1), with the exception of those samples labelled as containing oleic acid (7 samples). These had an assigned probability lower than 1 (Figure 6), which is why these samples were classified by the model as not-SF. When the non-targeted class was analysed, only 10 samples had an assigned probability value higher than 0. These samples corresponded mostly to pomace olive oil, for which the NSR RAMAN fingerprint should be different from the remaining olive oil fingerprints.

Finally, a SIMCA model as a soft modelling technique was performed. Three PCs were selected for the target and non-target classes, which explains the 93.4% and 30.5% of the cumulative variance, respectively.

Figure 7 shows the Cooman’s classification plot for the analysed samples. It can be seen that some samples appeared as not conclusive (SFO samples labelled as containing a high oleic acid content); as outliers (three SFO samples and most of the POO samples); and misclassified (a HOSFO sample assigned to the non-target class). Compared with the non-supervised patter recognition techniques, HCA and PCA, these samples appear grouped in the olive oil samples cluster or region.

Table 1, Table 2 and Table 3 show the quality parameters of these three models. In general, the SVM model produced the best result, followed by the kNN and SIMCA. Thus, it can be considered that the hard models correctly discriminate the samples according to the raw material. Nevertheless, the SIMCA soft model showed a better capacity to classify those sunflower oil samples with high oleic acid content, considering that they appear not only as sunflower but also as not-sunflower i.e., olive oil.

## 4. Conclusions

The chemometric/machine learning study of the spectroscopic instrumental fingerprints of edible vegetable oils (sunflower and olive oils), obtained using a highly versatile portable analyser based on Spatially Offset Raman Spectroscopy (SORS), showed that they are capable of distinguishing the analysed samples according to the original raw material used in the oil’s production.

Additionally, the obtained results are consistent with the revised bibliography. The normalised spatially resolved Raman fingerprints of different zones not only allowed for grouping of samples based on raw material, but also for differentiation within the groups based on oil types. Therefore, despite using a shorter wavelength range, the unsupervised techniques employed (HCA and PCA) demonstrated that the raw material, such as sunflower or olive oil, was the most influential attribute followed by the types (commercial categories) of each type of oil, both determined by the oleic acid content and the oleic/linoleic acid ratio in the analysed samples. This highlights the functional qualities of this type of sunflower oil and its importance in the sunflower oil market.

By applying supervised techniques, different models to discriminate (SVM and kNN) and classify (SIMCA) were obtained. Although reliable quality metrics of the models were satisfactory, as far as the authors are concerned, the best results were obtained in the SIMCA model. This soft classification model permitted not only the classification of samples according to the oil raw material (seed or fruit), but also the classification of those samples belonging to HOSFO as inconclusive due to their high oleic acid content similar to olive oil.

Finally, these results allow for the proposed combination of SORS and chemometric tools to be considered not only for use in quality control for routine analysis in the food industry, but also in other fields including the authentication of different types of edible oils, such as sunflower oil. Furthermore, instrumental fingerprints can be easily obtained through short-time analysis without any sample processing. This, combined with simple chemometric analysis, allows for the quality of genetically modified sunflower seeds to be highlighted for use in frying processes and baking.

## Figures and Tables

**Figure 1 foods-13-00183-f001:**
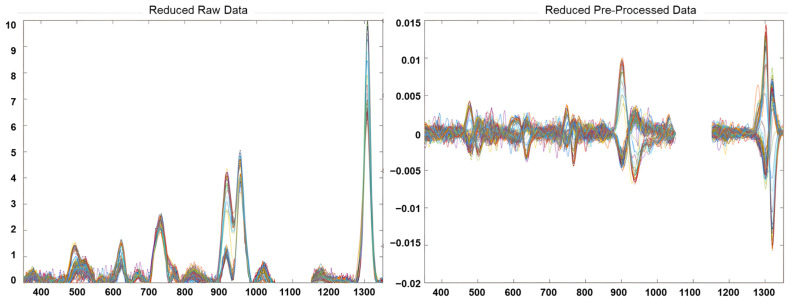
Reduced RAMAN fingerprints of edible vegetable oils before and after data pre-processing.

**Figure 2 foods-13-00183-f002:**
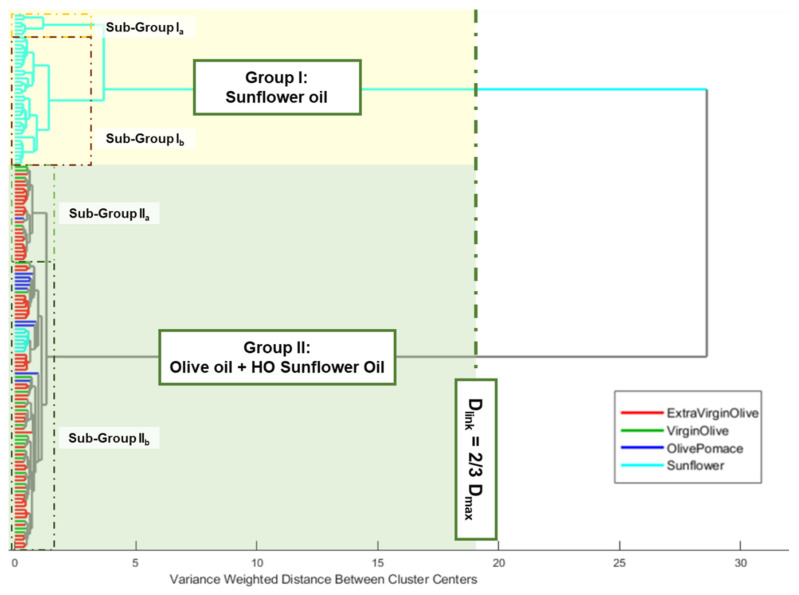
Dendrogram from HCA of analysed edible vegetable oils.

**Figure 3 foods-13-00183-f003:**
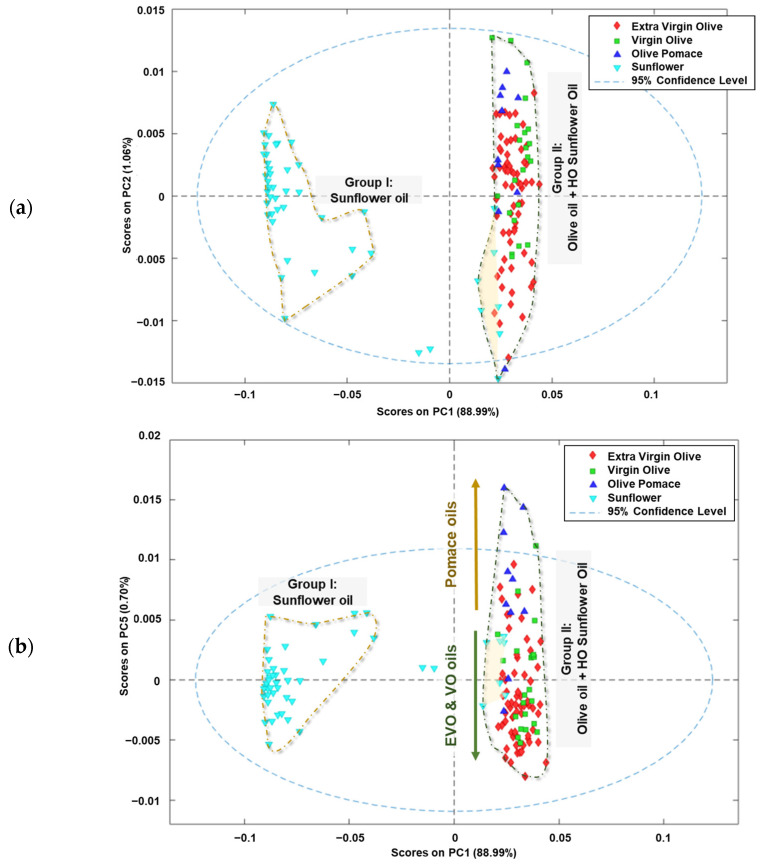
Score plots of analysed edible vegetable oils in: (**a**) plane PC2 vs. PC1, (**b**) plane PC5 vs. PC1 and (**c**) the 3D space defined by PC1 vs. PC2 vs. PC5.

**Figure 4 foods-13-00183-f004:**
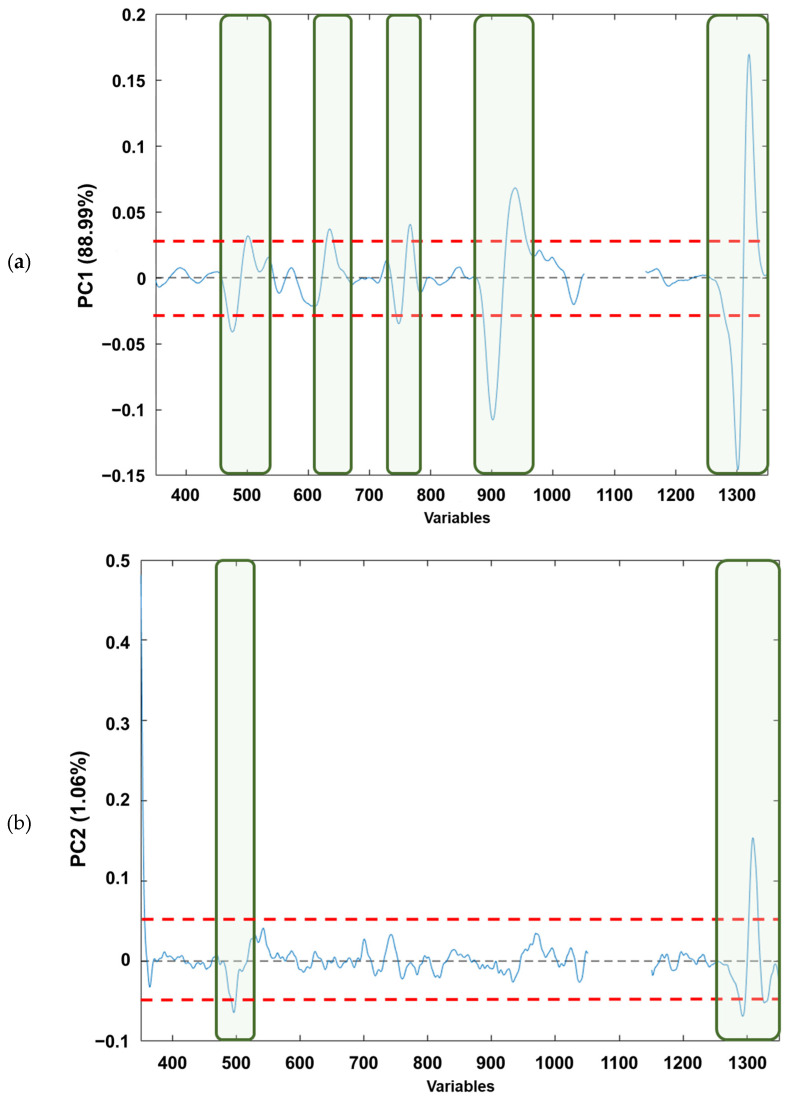
Loading plots from analysed edible vegetable oils in: (**a**) PC1, (**b**) PC2 and (**c**) PC5. The red dashed lines indicate the loading thresholds above which their contribution to natural sample grouping is considered.

**Figure 5 foods-13-00183-f005:**
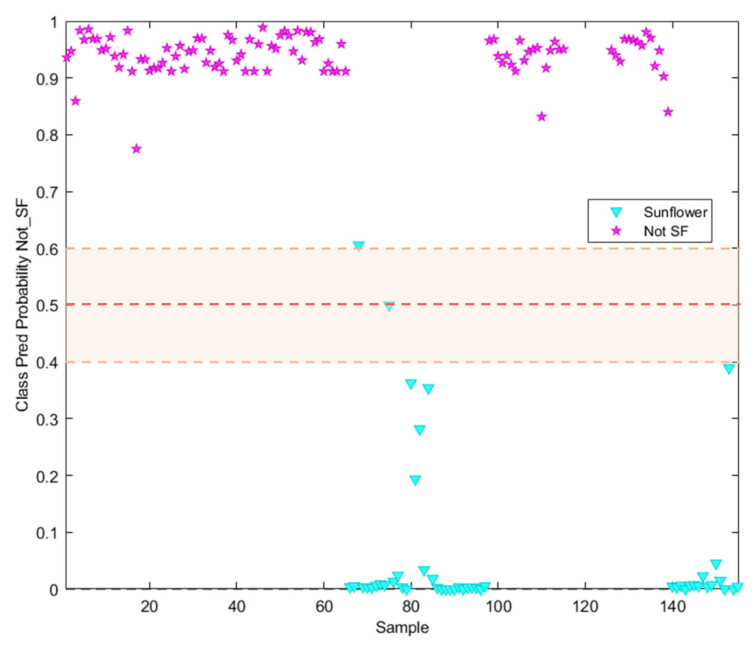
Probability prediction plot obtained from the SVM classification model. Dashed lines indicate class thresholds separating misclassified samples.

**Figure 6 foods-13-00183-f006:**
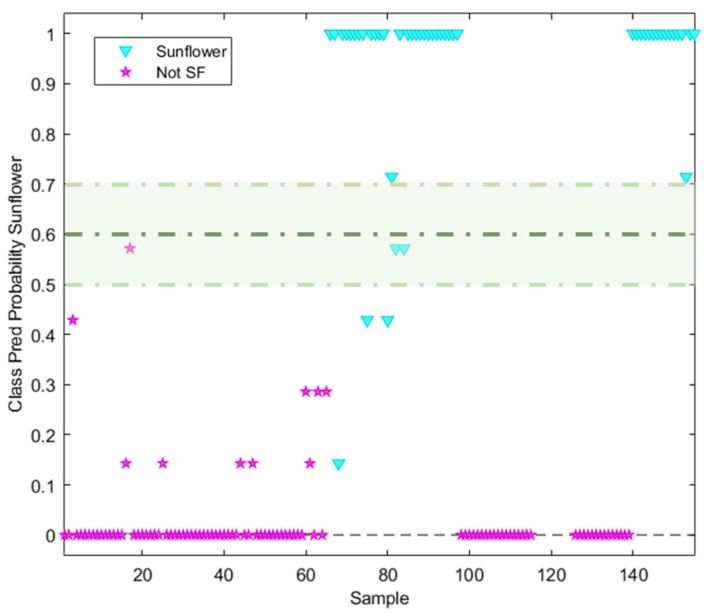
Probability prediction plot obtained from kNN (k = 7) classification model. Dashed lines indicate class thresholds separating misclassified samples.

**Figure 7 foods-13-00183-f007:**
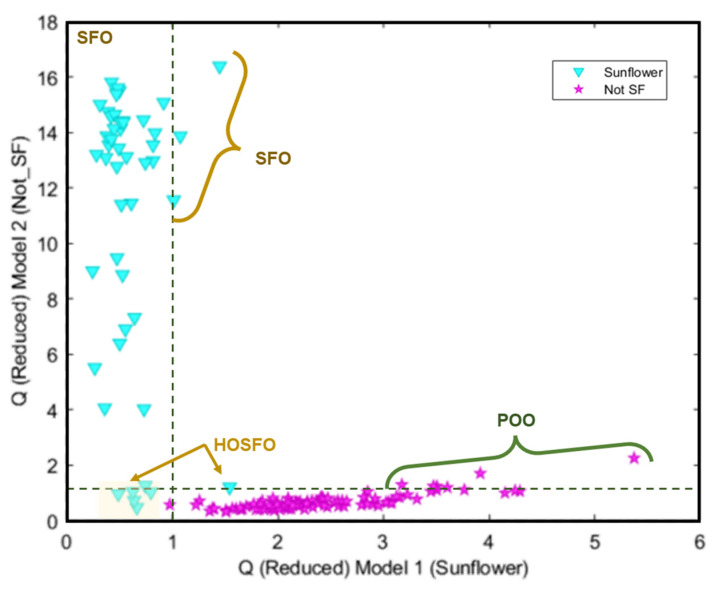
Cooman’s classification plots obtained for SIMCA model.

**Table 1 foods-13-00183-t001:** Summary of discrimination/classification performance metrics obtained for the SVM-DA one-input class model.

TARGET Class (TC): Sunflower Oil		
Features: X Block: [reduced RAMAN instrumental fingerprints] Y Block: [TC (sunflower oil, SFO); NTC (not sunflower oil, not SFO)] Preprocessing: 1st Derivative (order 2, window; 21 pt, tails: polyinterp) + Mean center Training Set: [97 × 902] Prediction Set: [48 × 902] See Confusion Table below		
Classification performance metrics	TC (SFO)	NTC (Not SFO)
Sensitivity (SENS -prediction stage)	0.98	1.00
Specificity (SPEC-prediction stage)	1.00	0.98
False positive rate (FPR)	0.00	0.02
False negative rate (FNR)	0.02	0.00
Positive predictive value (precision) (PPV)	1.00	0.99
Negative predictive value (NPV)	0.99	1.00
Youden index (YOUD)	0.98	0.98
F-measure (F)	0.99	0.99
Discriminant power (DP)	-	-
Efficiency (or accuracy) (EFFIC)	0.99	0.99
Misclassification rate (MR)	0.01	0.01
AUC (correctly classified rate)	0.99	0.99
Gini coefficient (Gini)	0.98	0.98
G-mean (GM)	0.99	0.99
Matthews correlation coefficient (MCC)	0.98	0.98
Chance agreement rate (CAR)	0.56	0.56
Chance error rate (CER)	0.44	0.44
Kappa coefficient (KAPPA)	0.98	0.98
**Confusion Table:**		
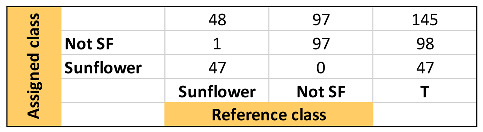		

Note. The hyphen “-” refers to metrics that cannot be determined since a division by zero is involved.

**Table 2 foods-13-00183-t002:** Summary of discrimination/classification performance metrics obtained for kNN-DA one-input class model.

TARGET Class (TC): Sunflower Oil		
Features: X Block: [reduced RAMAN instrumental fingerprints] Y Block: [TC (sunflower oil, SFO); NTC (not sunflower oil, not SFO)] Preprocessing: 1st Derivative (order 2, window; 21 pt, tails: polyinterp) + Mean center Training Set: [97 × 902] Prediction Set: [48 × 902] See Confusion Table below		
Classification performance metrics	TC (SFO)	NTC (Not SFO)
Sensitivity (SENS-prediction stage)	0.90	1.00
Specificity (SPEC-prediction stage)	1.00	0.90
False positive rate (FPR)	0.00	0.10
False negative rate (FNR)	0.10	0.00
Positive predictive value (precision) (PPV)	1.00	0.95
Negative predictive value (NPV)	0.95	1.00
Youden index (YOUD)	0.90	0.90
F-measure (F)	0.95	0.97
Discriminant power (DP)	-	-
Efficiency (or accuracy) (EFFIC)	0.97	0.97
Misclassification rate (MR)	0.03	0.03
AUC (correctly classified rate)	0.95	0.95
Gini coefficient (Gini)	0.90	0.90
G-mean (GM)	0.95	0.95
Matthews correlation coefficient (MCC)	0.92	0.92
Chance agreement rate (CAR)	0.57	0.57
Chance error rate (CER)	0.44	0.44
Kappa coefficient (KAPPA)	0.92	0.92
**Confusion Table:**		
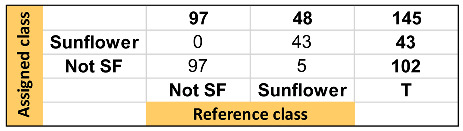		

Note. The hyphen “-” refers to metrics that cannot be determined since a division by zero is involved.

**Table 3 foods-13-00183-t003:** Summary of discrimination/classification performance metrics obtained for the SIMCA one-input class model.

TARGET Class (TC): Sunflower Oil		
Features: X Block: [reduced RAMAN instrumental fingerprints] Y Block: [TC (sunflower oil, SFO); NTC (not sunflower oil, not SFO)] Preprocessing: 1st Derivative (order 2, window; 21 pt, tails: polyinterp) + Mean center Training Set: [97 × 902] Prediction Set: [48 × 902] See Confusion Table below		
Classification performance metrics	TC (SFO)	NTC (Not SFO)
Sensitivity (SENS -prediction stage)	0.93	0.81
Specificity (SPEC-prediction stage)	0.81	0.93
False positive rate (FPR)	0.19	0.07
False negative rate (FNR)	0.07	0.19
Positive predictive value (precision) (PPV)	0.99	1.00
Negative predictive value (NPV)	1.00	0.99
Youden index (YOUD)	0.74	0.74
F-measure (F)	0.96	0.90
Discriminant power (DP)	0.96	0.96
Efficiency (or accuracy) (EFFIC)	0.89	0.89
Misclassification rate (MR)	0.11	0.11
AUC (correctly classified rate)	0.87	0.87
Gini coefficient (Gini)	0.74	0.74
G-mean (GM)	0.87	0.87
Matthews correlation coefficient (MCC)	0.86	0.86
Chance agreement rate (CAR)	0.51	0.51
Chance error rate (CER)	0.44	0.44
Kappa coefficient (KAPPA)	0.78	0.78
**Confusion Table:**		
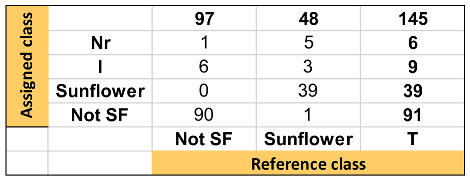		

Note. Nr: not recognised; I: inconclusive.

## Data Availability

The corresponding authors can provide the data presented in this study upon request. The data is not publicly available at this time but will be uploaded to the institutional repository (UGR Repository) after the article is published.

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
