# Peer review of "Discrimination/Classification of Edible Vegetable Oils from Raman Spatially Solved Fingerprints Obtained on Portable Instrumentation"

_foods, 2024, doi:10.3390/foods13020183_

Round 1
Reviewer 1 Report
Comments and Suggestions for Authors
This article applied portable Raman device to detect olive oil and sun flower oil. Various machine learning models were established. The results showed good discrimination power. There are some commends:
1. There are two oils be classified in the study. Please add the reason why choose sun flower oil and olive oil. Is they similar or commonly mixed.
2. The sample were all from commercial sources. How authors identify their authenticity.
3. the result and discussion part need references. for example, line 198-213 dressed the vibration and bonds, please add some references and discuss your finding to similar studies using Raman for oil detection.
4. Same issue, for PCA, please add reference in the discussion. There are plenty of literature talking the adulteration of olive oil by Raman.
5. in 3.2, kNN and SIMCA model were used to discriminated sunflower oil and not SF. What will it be if use olive oil and not olive oil.
6. Did author normalize data before running models? If so please added in data treatment.
Comments on the Quality of English Language
English is fine.
Author Response
The authors would like to thank not only for the work developed by the reviewer but also for the comments and suggestion which without doubt will improve the quality of the manuscript.
Please find attached the point by point responses in the word document.

Reviewer 2 Report
Comments and Suggestions for Authors
The similarity with these 2 sources needs to be reduced.
1)link.springer.com/article/10.1007/s11947-023-03039-8?code=4cccca93-0022-40d2-baec-ff473e47b517&error=cookies_not_supported
2)
doi.org/10.1016/j.jfca.2022.104904
Extensive English editing required.
11 friendly-environmental --> remove this word, not added value. Also, it should be environmentally friendly
15 reliable -> reliably
16, 17: HCA and PCA as unsupervised pattern recognition techniques but also SVM, kNN and SIMCA --> put acronyms in brackets
28 In the European Union, represents -->noun missing in the sentence
Comments on the Quality of English Language
Several formulations that need to be improved for better readability.
Author Response
The authors would like to thank not only for the work developed by the reviewer but also for the comments and suggestion which without doubt will improve the quality of the manuscript.
Please find attached the point by point responses in the word document.

Reviewer #2
Comments and Suggestions for Authors:
The authors would like to thank not only for the work developed by the reviewer but also for the comments and suggestion which without doubt will improve the quality of the manuscript.
- The similarity with these 2 sources needs to be reduced:
- springer.com/article/10.1007/s11947-023-03039-8?code=4cccca93-0022-40d2-baec-ff473e47b517&error=cookies_not_supported
- org/10.1016/j.jfca.2022.104904
It has done. Despite this, the similarity arises from previous research developed by some of the authors and includes the same chemometric approach. Therefore, as far as the authors are concerned, this similarity should not be considered plagiarism.
Additionally, to avoid any plagiarism problems, the main issues have been rewritten.
- Extensive English editing required.
Attending your suggestion, an English edition has been done.
- Line 11 friendly-environmental à remove this word, not added value. Also, it should be environmentally friendly
In this case, the author consider that it is important to point out the greenish of this kind of instrumentation, so the correction suggested has been done and the concept have been kept in the text.
- 15 reliable -> reliably
Done.
Reviewer 3 Report
Comments and Suggestions for Authors
In this manuscript, an innovative approach was developed for the discrimination and classification of edible vegetable oils using Spatially Offset Raman Spectroscopy (SORS). By integrating SORS with advanced pattern recognition techniques, Raman fingerprints of the oils were rapidly obtained. The Raman results provided the detailed molecular information for discrimination and classification purposes. The environmentally friendly and economical technique developed in this study to classify oils could be an alternative to traditional approaches. Although this study is an interesting work on rapid analytical techniques in the field of food, I think it can be published if the following corrections should be made.
- Raman has been used for many years in the classification of oils. The advantage of using chemometrics/machine learning in this study should be explained in detail.
- It is not possible to read the y-axis of Figure 2, please revise it.
- Has validation or comparison been done with any other approach? Have at least repeat experiments been performed? How did the authors come to the conclusion that the developed approach is a reliable technique as stated in the manuscript?
Author Response
The authors would like to thank not only for the work developed by the reviewer but also for the comments and suggestion which without doubt will improve the quality of the manuscript.
Please find attached the point by point responses in the word document.

Reviewer #3
Comments and Suggestions for Authors
In this manuscript, an innovative approach was developed for the discrimination and classification of edible vegetable oils using Spatially Offset Raman Spectroscopy (SORS). By integrating SORS with advanced pattern recognition techniques, Raman fingerprints of the oils were rapidly obtained. The Raman results provided the detailed molecular information for discrimination and classification purposes. The environmentally friendly and economical technique developed in this study to classify oils could be an alternative to traditional approaches. Although this study is an interesting work on rapid analytical techniques in the field of food, I think it can be published if the following corrections should be made.
The authors would like to thank not only for the work developed by the reviewer but also for the comments and suggestion which without doubt will improve the quality of the manuscript.
- Raman has been used for many years in the classification of oils. The advantage of using chemometrics/machine learning in this study should be explained in detail.
Although the fingerprinting methodology needs of powerful data analysis provided by chemometric machine learning and it is well known for chemometrician and machine learning users, a new paragraph has been introduced in the revised marked version in lines 93 to 98.
- It is not possible to read the y-axis of Figure 2, please revise it.
The authors have removed the text referring to the y-axis as they consider it to be an internal quality requirement. They believe that the type of edible vegetable oil can be associated with the color described in legend box, making the texted information unnecessary.
- Has validation or comparison been done with any other approach? Have at least repeat experiments been performed? How did the authors come to the conclusion that the developed approach is a reliable technique as stated in the manuscript?
In all cases, a validation of all chemometric models has been carried out and no other approach has been used. Replicates of the measurement were randomly performed on each type of oil for analytical validation purposes. However, these replicates were not included in the chemometric models as they could potentially distort the results obtained from the original instrumental fingerprint matrix. Regarding the conclusion, the authors state that the analysis of oil samples using SORS-based portable equipment is adequate. This is supported by the chemometric study from different techniques. Additionally, the fact that different chemometric methods based on different principles reach similar conclusions can be considered as a validation in itself.
Round 2
Reviewer 1 Report
Comments and Suggestions for Authors
Significant improvement was found in this article. In the conclusion, the results were repeatly dressed. The addition of comments or applications in the conclusion may extend the depth of this paper.
Comments on the Quality of English Language
There is only minor English problem. Some parts are little hard to read.
Author Response
The authors are in accordance with these suggestions and thus the Conclusion section has been modified to highlight the goodness of the proposed methodology and their importance in food industry.

Comments and Suggestions for Authors
Significant improvement was found in this article. In the conclusion, the results were repeatly dressed. The addition of comments or applications in the conclusion may extend the depth of this paper.
The authors are in accordance with these suggestions and thus the Conclusion section has been modified to highlight the goodness of the proposed methodology and their importance in food industry.
Reviewer 3 Report
Comments and Suggestions for Authors
Edits and comments made by the authors are sufficient for me. In my opinion, this version of the article is suitable for publication.
Author Response
thank you